# Identification of key odorants in honeysuckle by headspace-solid phase microextraction and solvent-assisted flavour evaporation with gas chromatography-mass spectrometry and gas chromatograph-olfactometry in combination with chemometrics

Keran Su[1], Xin Zhang[2], Shao Quan Liu[1]*, LiHui Jia[2], Benjamin Lassabliere[3], Kim Huey Ee[3], Aileen Pua[1], Rui Min Vivian Goh[1], Jingcan Sun[3], Bin Yu[3]*, XiaoXue Hu[4]*

1 Department of Food Science and Technology, National University of Singapore, Singapore, Singapore,
2 Key Laboratory for Green Chemical Process of Ministry of Education, School of Chemistry and Environmental Engineering, Wuhan Institute of Technology, Wuhan East Lake New Technology Development Zone, Wuhan, China, 3 Mane SEA Pte Ltd, Singapore, Singapore, 4 Hubei Provincial Hospital of Traditional Chinese Medicine, Wuhan, China

* huxiaoxue@hbhtcm.com (XXH); gsgpbiy@gmail.com (BY); fstlsq@nus.edu.sg (SQL)

## Abstract

At present, the identification of honeysuckle aroma depends on experienced tasters, which results in inconsistencies due to human error. The key odorants have the potential to distinguish the different species and evaluate the quality of honeysuckle. Hence, in this study, a more scientific approach was applied to distinguish various honeysuckles. The volatile compounds of different species and parts of honeysuckle were separately extracted by headspace-solid phase microextraction (HS-SPME) and solvent assisted flavor evaporation (SAFE). Compounds with greater volatility such as aldehydes, limonene, γ-terpinene, and terpinolene were preferentially extracted by HS-SPME. As a complementary extraction method to HS-SPME, SAFE was found to recover comparatively more polar compounds such as eugenol, decanoic acid, and vanillin. Subsequently, key odorants with the highest flavour dilution (FD) factors were detected by aroma extract dilution analysis (AEDA). These were benzaldehyde, 4-ethylphenol, decanoic acid, vanillin, 3-methyl-2-butenal, and β-ionone in honeysuckle flowers and γ-octalactone, 4-ethyl phenol, and vanillin in honeysuckle stem. Finally, principal component analysis (PCA) was conducted to analyze not only the key odorants of species and parts of honeysuckle but also their different origins. The results of PCA suggested that the species of honeysuckle contributed much more to variations in aroma rather than their origins. In conclusion, the application of the key odorants combined with PCA was demonstrated as a valid approach to differentiate species, origins, and parts of honeysuckle.

**Data Availability Statement:** All relevant data are within the manuscript and its Supporting Information files.

**Funding:** Mane SEA Pte Ltd provided support in the form of salaries for BL, KHE, JS, and BY and research materials. The specific roles of each authors are articulated in the 'author contributions' section.

**Competing interests:** The authors declare that no competing interests exist. Mane SEA Pte Ltd does not alter our adherence to PLOS ONE policies on sharing data and materials.

## Introduction

Honeysuckle is a kind of herbal plant with many species and widely used and grown in northern hemisphere countries [1]. It has been consumed as traditional medicine and tea beverage for thousands of years [2]. In addition to the treatment of pneumonia and respiratory infections, and the improvement of consumers' health, honeysuckle possesses anti-inflammatory, antioxidant, and antiviral properties [3,4]. For this reason, it has been gradually added into foods and beverages to improve product value and to provide adjuvant therapy [5,6]. Recently, the aroma of herbal plants in food products has also increasingly gained popularity and consumer acceptance around the world [7]. Key aroma profiles helps to distinguish the different species and determines the quality of honeysuckle. At present, the quality of honeysuckle is evaluated by professional tasters that causes inconsistent and inaccurate results [8]. Therefore, a systematic study of the volatile compositions of different species, parts and origins of honeysuckle was conducted in this study.

The selection of an appropriate extraction method is a prerequisite to study volatile compounds in honeysuckle. The complex plant matrix requires the development of various sample preparation approaches to address the issue of sample complexity [9]. Some solvent extraction methods operate under high temperatures and use toxic organic solvents that are prone to thermal reactions and cause adverse effects to environment. In comparison, headspace-solid phase microextraction (HS-SPME) is a relatively low-temperature, solvent-free, and rapid approach [10]. For this reason, it is a popular method applied in the food matrix of natural materials [11,12]. HS-SPME is suitable for the isolation of highly volatile compounds. In comparison to HS-SPME, solvent-assisted flavour evaporation (SAFE) is able to extract less volatile and a greater range of polar compounds from complex matrices [13]. By applying high vacuum and low-temperature conditions, SAFE avoids aroma modification during the extraction of heat-liable aroma compounds from natural materials [14]. Besides, SAFE reduces the number of side reactions that may occur during aroma isolation, such as enzymatic degradation [15]. Hence, these two extraction methods are suitable for isolating volatile compounds from natural materials.

The potency of the aroma is often not proportionate to its peak area detected by chromatography-mass spectrometry (GC-MS) because many high abundance compounds may have a high odor threshold and thus do not serve as key odorants in the overall matrix [16]. Hence, it was difficult to distinguish between the aromas of different honeysuckle species using the volatile characteristics alone. As well known, aroma extract dilution analysis (AEDA) has been widely used to fill this gap by suggesting the contribution of each compound through the flavour dilution (FD)-factor [17]. The key odorants detected by AEDA may then be used to better describe the characteristic aroma of honeysuckle, and has the added potential to distinguish different species of honeysuckle.

Therefore, the objective of this study was to extract the volatile compounds in various honeysuckles using HS-SPME in conjunction with SAFE, both coupled to GC-MS/FID. Furthermore, their key odorants were identified using AEDA. Lastly, principal component analysis (PCA) was performed to show the differences in the aromatic profiles of honeysuckle samples from different species, origins, and parts of the plants.

## Materials and methods

### Sample preparation

Two main species of honeysuckle flowers are *Lonicera japonica Flos* (Jinyinhua) and *Lonicera Flos* (Shanyinhua) [18]. The stems of honeysuckle are also commonly used as medicine [6]. All

of the flowers and stems of honeysuckle belong to the family of *Caprifoliaceae* [19] and contain similar chemical constituents [6], but they cannot be used interchangeably. With the help of Hubei Provincial Hospital of Traditional Chinese Medicine, six common honeysuckle samples with permit number 20160077 were purposefully selected for this study. These are flowers of honeysuckle: *Lonicera japonica Flos* (LJF, Pingyi County, Shandong province, China) and *Lonicera Flos* (LF, Longshan County, Hunan province, China), stems of *Lonicera japonica*: *Lonicera japonica Caulis* (LJC, Pingyi County, Shandong province, China), and stems of *Lonicera*: *Lonicera Caulis* (LC, Longshan County, Hunan province, China). In addition, LJF from Fengqiu County, Henan province, China and LF from Enshi County, Hubei province, China were also complemented for PCA to compare the differences between origins. All samples were packed in heat-seal vacuum aluminium foil bags for every 100 g portion sand stored in a 5˚C refrigerator. The aluminium foil bag can isolate the external humidity, and all samples were used within one week after repacking.

## HS-SPME procedure

The HS-SPME procedure was conducted as described in our previous publication [20]. Two different sample preparation methods, namely dry and brewed, were applied in this study. In the dry method, 0.200 g of LJF (Shandong or Henan), LF (Hunan or Hubei), LC, or LJC was weighed (d = 0.001 g and max 620 g, Mettler Toledo, America) and added directly in a glass headspace vial with a PTFE-coated silicone septum (Agilent, California, USA). The brewed honeysuckle was prepared by adding 20.000 g of honeysuckle in 200.000 g of 80˚C Ultrapure water (Human Corporation Arioso Power Series, Seoul, South Korea). The optimal brewing conditions were determined to be 80˚C and 30 min (results not shown here). After brewing, the mixture was filtered through a metal sieve and chilled in an ice bath for 10 min. The cooled filtrate (2.000 g) was then transferred into a headspace vial. A Supelco 85 μm Carboxen/Polydimethylsiloxane (CAR/PDMS) fibre (Pennsylvania, USA) was used to extract volatiles from headspace vial at designated extraction temperatures (40, 60, and 80˚C) and extraction times (5, 30, and 60 min). Finally, the fibre was inserted in the GC injector for the subsequent GC analysis.

## SAFE procedure

Brewed honeysuckle was prepared similarly as described above but in this case 60.000 g of sample was added in 600.0 g of 80˚C Ultrapure water was weighed by the balance (d = 0.1 g and max 8200). After brewing at 80˚C for 30 min, the mixture was filtered through a metal sieve and chilled in an ice bath for 10 min. The chilled filtrate (300 mL) was then transferred into a 500 mL beaker. An internal standard solution (0.060 g) was added to the filtrate and stirred for 5 min. The internal standard solution was prepared as follows: 0.300 g butyl butyryl lactate (VWR, Pennyslvania, USA), which was found not to coelute with other volatile compounds in honeysuckle in both MS and FID chromatograms, was added in 250 mL volumetric flask, then dilute to volume with ethanol (VWR, Pennyslvania, USA). The formula for calculation of the concentration of each compound was shown in Eq (1). The extraction conditions of SAFE was adapted from our previous work, with minor modifications [21]. The extraction conditions of SAFE (Glasbläserei Bahr, Germany) were $5 \times 10^{-6}$ mbar, the pressure was maintained by Edwards PFPE RV3 rotary vane pump (West Sussex, UK) and Edwards Diffstak diffusion pump (West Sussex, UK), and the water bath temperature was 40˚C.

$$\frac{Peak\ area\ of\ internal\ standard}{concentration\ of\ internal\ standard} = \frac{Peak\ area\ of\ each\ compound}{concentration\ of\ corresponding\ compound} \quad (1)$$

## GC-MS/FID analysis

Analytes quantitation and identification analysis were conducted by Agilent 7890B GC equipped with a flame ionisation detector (FID) and Agilent 5977B mass selective detector (MSD) (California, USA). CTC CombiPAL autosampler (Zwingen, Switzerland) was employed to insert samples into GC injector. GC column was Agilent HP-INNOWax column (60 m × 250 μm × 0.25 μm) (Woodbridge, USA). The GC conditions used were injector temperature 250°C; splitless mode; helium carrier gas; column flow rate 1.2 ml/min; FID temperature was 300°C; EI mode was 70 eV. The temperature gradient in the GC oven was 50°C for 5 minutes, increased to 240°C at rate of 5°C/min, and held at 240°C for 40 minutes. In-house and NIST 14 MS libraries (Connecticut, USA) were applied to identify the eluted compounds by matching the mass spectra, and further confirmed with the linear retention indices (LRI) of in-house standard compounds. LRI values of the HP-INNOWax column were calculated by a mixture of Supelco C7-40 alkanes (VWR, Pennsylvania, USA). Alkane standard and pure standards were run under the same conditions with the sample. All experiments were carried out in triplicate, and the results were reported as mean values of the peak area of each compound extracted by HS-SPME and mean values of concentration of each compound extracted by SAFE.

## AEDA

For the AEDA analysis, the SAFE concentrates of LJF, LF and LJC were diluted stepwise with dichloromethane (VWR; Pennyslvania, USA) at $5^n$ (n = 0, 1, 2, 3, and so on). Each dilution was sniffed and evaluated via GC-O by four experienced flavourists (2 male and 2 female; aged 26–50). Gas chromatograph-olfactometry (GC-O) conditions were referred from our previous study [22]. All GC-O panellists had experience in conducting GC-O analyses before this study. The odor of each compound was described by these panellists. The strength of each compound was represented by their flavour dilution factor (FD-factor), which was the highest dilution at which nothing could be smelled at the GC-O port. Compounds corresponding to each odor were identified by injection of pure standards under the same GC-O condition.,

## Data processing and statistical analysis

Student's *t*-test was performed to test the significance level of the difference between samples at two different conditions. Data of key aroma compounds obtained from AEDA was processed by PCA in the R programming language. The data was processed and transformed with data.table and magrittr packages and visualized them with ggplot2, ggbiplot, and patchwork packages.

# Results and discussion

## The selection of HS-SPME extraction conditions

It is critical to select the extraction conditions (temperature and time) for natural materials. That is because high temperature and long-time extraction could destroy their volatile compounds, while lower temperatures and shorter extraction times could lead to poor sensitivity and recovery [23,24]. Therefore, different extraction temperatures (40, 60, and 80°C) and extraction times (5, 30, and 60 min) were applied to study the effects of extraction conditions on the release of volatile compounds from honeysuckle (both dry and brewed).

As reported in the literature [25–28], hexanal, *cis*-3-hexenol, acetic acid, benzyl alcohol, and *β*-ionone were important compounds in honeysuckle. Hence, they were chosen as indicators for the effect of extraction temperatures on both dry and brewed LJF. In Fig 1(A), the peak

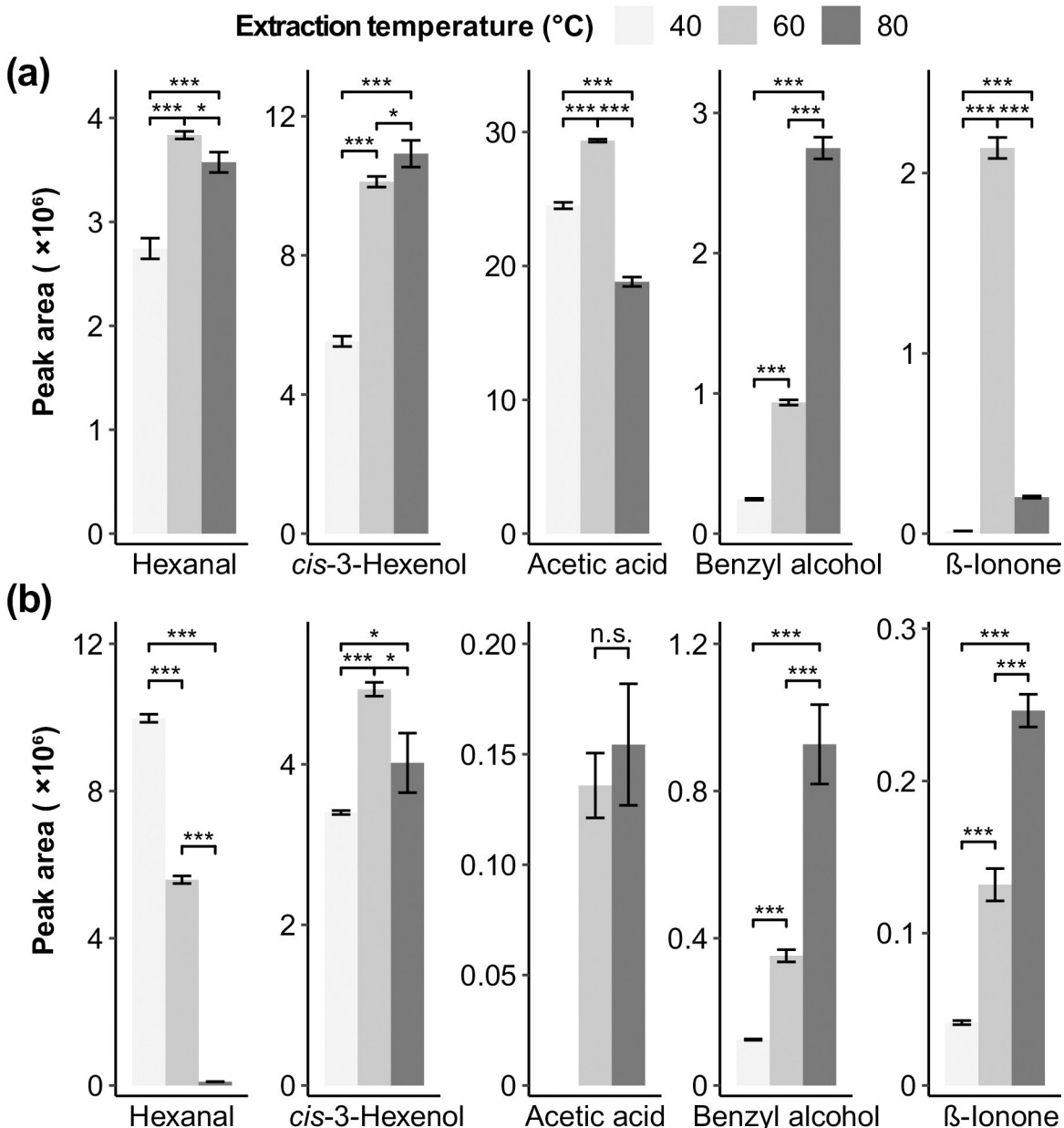

**Fig 1.** Extraction temperature profiles of volatile compounds in *Lonicera japonica Flos* extracted by using HS-SPME for 30 min: (a) Dry *Lonicera japonica Flos*; (b) brewed *Lonicera japonica Flos*. Star signs are given according to *t*-test *p* values ($^*$: $0.01{\leq}p{<}0.05$, low significance; $^{**}$: $0.001{\leq}p{<}0.01$, medium significance; $^{***}$: $p{<}0.001$, high significance).

area of hexanal, acetic acid, and β-ionone increased with the increase of temperature but decreased after 60˚C. Other aldehydes (e.g. 2-butenal, heptanal, *trans*-2-hexenal, and benzaldehyde) exhibited a similar trend. For *cis*-3-hexenol and benzyl alcohol, an increase in temperature resulted in the higher peak areas, which was observed for other alcohols such as geraniol, octanol, and 2-phenylethyl alcohol. Hence, alcohols were easier to be detected at the higher temperatures in dry honeysuckle.

The peak areas of the selected compounds after brewing are shown in Fig 1(B). Hexanal and other aldehydes such as 2-butenal, heptanal, *trans*-2-hexenal, and benzaldehyde showed

an inverse trend compared to dry LJF. Hence, aldehydes were a little bit harder to be detected at a higher temperature in brewed honeysuckle. When the temperature was gradually increased from 40°C to 80°C, the peak areas of the most alcohol compounds showed high significant increase, except *cis*-3-hexenol, which reached the greatest peak area at 60°C. *cis*-3-Hexenol appeared to stop rising at 60°C, accompanied by increments in geraniol and β-ionone in samples after brewing, which is similar to the previous study [29], which may be the result of thermally-related reactions. The peak areas of acetic acid and *β*-ionone had the same trends as the peak areas increase when the extraction temperature was above 60°C. Compared to dry honeysuckle extraction, acetic acid was more difficult to be isolated in brewed honeysuckle.

Fig 2 shows the peak areas of major volatile compounds in dry and brewed LJF at various extraction times at 80°C. In Fig 2(A), hexanal, *cis*-3-hexenol, and acetic acid were found to have the highest peak areas at 30 min, while benzyl alcohol and *β*-ionone recorded their respective highest peak areas at 60 min in dry LJF. For brewed LJF (Fig 2(B)), an extraction time of 30 min resulted in the greatest peak areas for *cis*-3-hexenol, acetic acid, and *β*-ionone. The peak area of benzyl alcohol was the highest with a 60 min extraction time, similar to dry LJF. Hexanal exhibited a unique trend. When the extraction time increased from 5 min to 30 min, the peak area of hexanal significantly decreased ($p<0.001$), and as the extraction time increased from 30 min to 60 min, the peak area of hexanal significantly increased ($p<0.001$). This phenomenon may be due to the hydration of hexanal and it could be fully extracted at low temperature [21].

The trends of each compound in dry LJC and LJC after brewing were similar to dry LJF and LJF after brewing. Hence, the results of the LJC were not shown in this study. Most of the volatile compounds in dry and brewed honeysuckle generally increased with the increase of temperature and had the greatest peak area at 30 min. Therefore, 80°C and 30 min were chosen as the extraction conditions both in dry and brewed samples for further analysis.

## Volatile profiles of different species and parts of honeysuckle by HS-SPME

Following the selection of HS-SPME extraction conditions, dry and brewed LJF, LF, LJC, and LC were analysed by GC-MS/FID under HS-SPME conditions of 80°C and 30 min. One hundred and two compounds were tentatively identified using HS-SPME-GC-MS and shown in Fig 3. Higher volatile compounds were preferentially found, such as some aldehydes, limonene, *γ*-terpinene and terpinolene. The results are shown in S1 Table. The most abundant compounds in dry LJF were 2-butenal, hexanal, hexanol, *cis*-3-hexenol, acetic acid, benzaldehyde, octanol, benzyl alcohol, and 2-phenylethyl alcohol, etc., and these compounds have also been reported by Wang et al. [30]. The main compounds in brewed LJF were almost the same as those in dry LJF. However, the abundances of the same compounds were different in brewed and dry LJF. For example, the peak areas of aldehydes and acetic acid in brewed LJF were lower while alcohol compounds were greater when compared to dry LJF. The abundance differences were likely due to the different sample matrix. The changes of volatile compounds of LF were similar to LJF. Although hexadecenoic acid and octadecanoic acid were reported to be the quantitatively highest compounds in *Lonicera Flos* [31], they were not detected in our study. This might be due to the different extraction methods, such as microwave, hydro distillation, and ultrasound, it used. These methods cause thermal reactions or decomposition during extraction. Some terpenes (e.g. *α*-copaene, *α*-cubebene, *γ*-muurolene, and *α*-muurolene) were detected in dry LJC and LC but not in brewed stem and flower samples. Until now, there have been few studies on volatile compounds of honeysuckle, but they only focused on one species such as LJF or LF [25,30]. Few studies have been conducted to differentiate the volatile compounds in honeysuckle flower of two species, or in the different parts of a single species

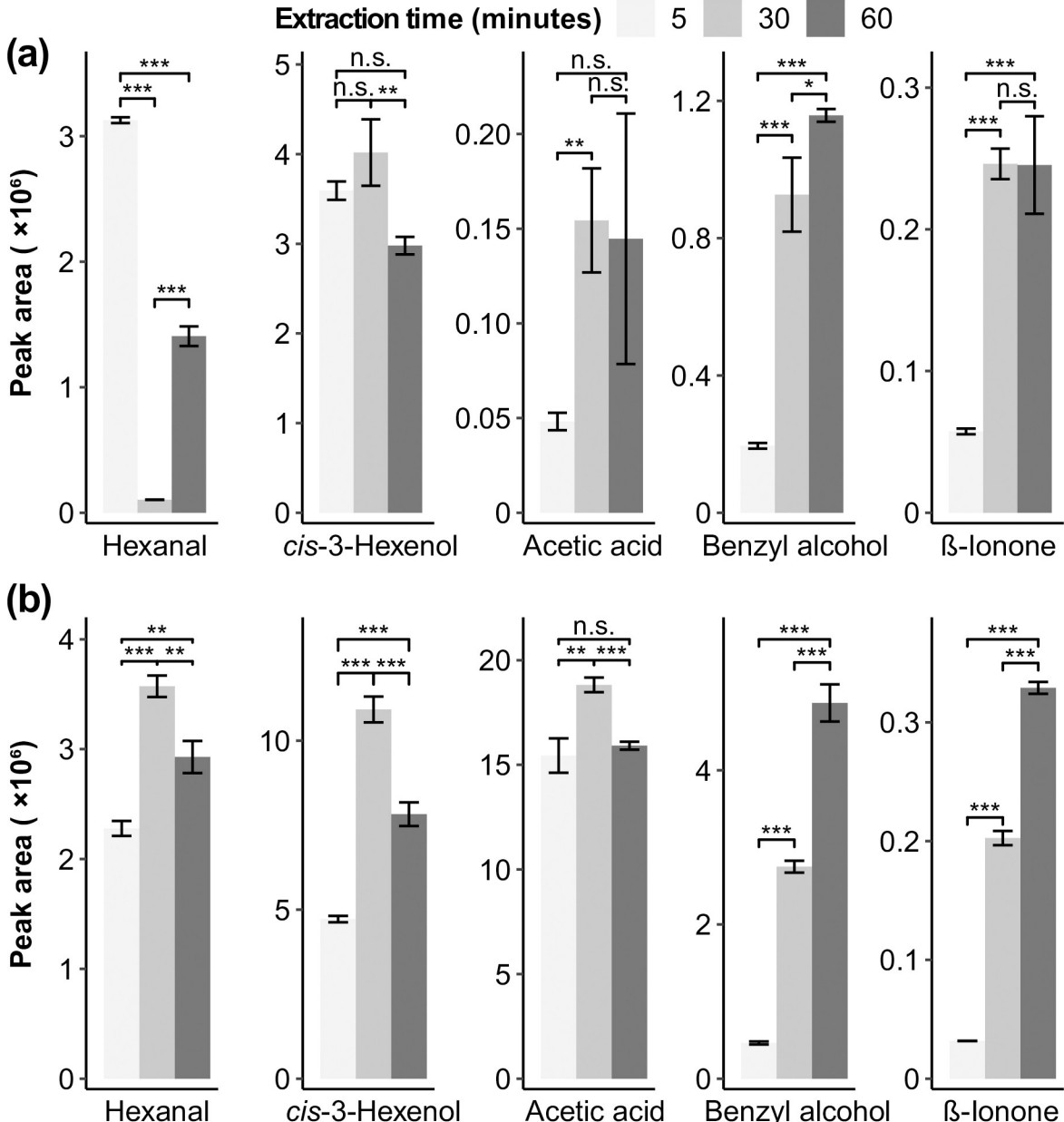

**Fig 2.** Extraction time profiles of volatile compounds in *Lonicera japonica Flos* extracted by using HS-SPME at 80°C: (a) Dry *Lonicera japonica Flos*; (b) Brewed *Lonicera japonica Flos*. Star signs are given according to *t*-test *p* values (*: $0.01 \leq p < 0.05$, low significance; **: $0.001 \leq p < 0.01$, medium significance; ***: $p < 0.001$, high significance).

[6,32]. Hence, volatile compounds in various honeysuckles and different status may provide different aroma spectrum when they were added in food products.

## Volatile profiles of different species and parts of honeysuckle extracted by SAFE

LJF, LF, LJC, and LC were isolated at 40°C water bath by SAFE. S2 Table listed one hundred and eight compounds across all honeysuckle species, such as hexanal, 1-octen-3-ol,

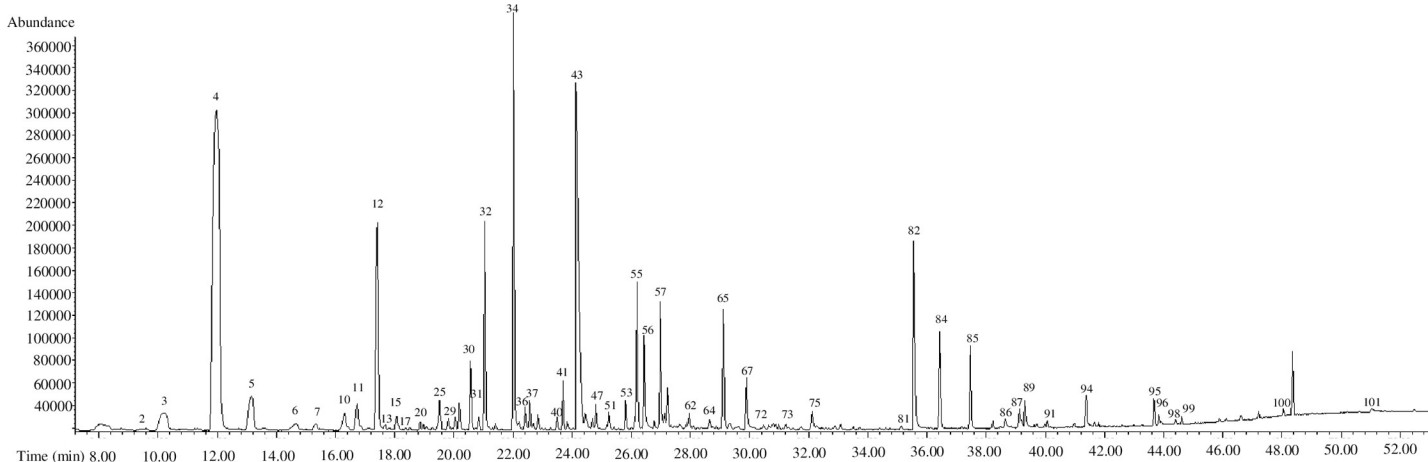

**Fig 3. GC-MS/FID chromatogram of _Lonicera japonica Flos_ (Shandong) extracted by HS-SPME at the extraction conditions of 80˚C and 30 min.** The numbers of each peak correspond to the compounds in S1 Table.

benzaldehyde, and eugenol, and these compounds were reported by literature [32]. SAFE was a complementary extraction method to HS-SPME due to its success in extracting higher polar compounds, higher molecular weight, and less volatile compounds such as eugenol, decanoic acid, and vanillin. SAFE is a modified distillation technique conducted under low-temperature and high-vacuum conditions, which aims to reduce the degradation of heat-sensitive volatiles and generation of artifacts as compared to other extraction methods like simultaneous distillation extraction (SDE) and hydro-distillation. When the extraction is performed using SDE, many linalool derivatives such as linalool oxides (furanoid and pyranoid) [25] as well as esters (bornyl actate, methyl hexadecanoate and methyl linolenate) [33] were detected. All these volatiles might be the result of thermal reactions during the SDE extraction process. Volatile compounds with lower boiling points such as pentanal, hexanal, trans-2-pentenal, and heptanal could not be detected by hydro-distillation [34] due to volatilisation during extraction. Therefore, the volatile compounds isolated by SAFE minimises the modification of volatiles in honeysuckle to obtain a more representative aroma extract.

## Key odorants of honeysuckle

Thirty-six odorants were identified in LJF, LF, and LJC (Table 1). These odorants with the highest FD-factors had sweet, floral, fruity, burnt, phenolic, and vanilla-like odor qualities. Benzaldehyde (bitter almond, burnt sugar; FD-factor: 625), 4-ethyl phenol (phenolic, smoky; FD-factor: 625), decanoic acid (sour, citrus; FD-factor: 625) and vanillin (vanilla, sweet; FD-factor: 625) were detected at the highest FD-factors in LJF. In LF, 3-methyl-2-butenal (fruity, sweet; FD-factor: 625) and $\beta$-ionone (seaweed, floral; FD-factor: 625) were the most potent odorants by FD-factor. The highest FD-factors in LJC was 125, and these odorants were $\gamma$-octalactone (sweet, waxy; FD-factor: 125), 4-ethyl phenol (phenolic, smoky; FD-factor: 125) and vanillin (vanilla, sweet; FD-factor: 125).

Some odorants were found to be unique to a particular honeysuckle species. For example, decanoic acid (sour, citrus; FD-factor: 625) showed the highest FD-factors in LJF, but it was not detected in LJC and LF. 1-Penten-3-ol (horseradish, green; FD-factor: 125), 3-methyl-2-butenal (fruity, sweet; FD-factor: 625), _trans_-2-octenal (waxy, green; FD-factor: 125) had high FD-factors in LF, but were not found to be key odorants in LJF and LJC. On the other hand, $\gamma$-octalactone (sweet, waxy; FD-factor: 125) had the highest FD-factor in LJC but it was

**Table 1. Key odorants (flavour dilution factor ≥ 1) with their respective flavour dilution factors in SAFE extracts of *Lonicera japonica Flos*, *Lonicera Flos* and *Lonicera japonica Caulis*.** Olfactometric detection was conducted by four flavourists.

| No. | Compound | Expt. LRI[†] | Ref. LRI[‡] | Odor quality[§] | Flavour dilution factor[‖] | | | Identification[⊥] |
|---|---|---|---|---|---|---|---|---|
| | | | | | Lonicera japonica flos | Lonicera flos | Lonicera japonica Caulis | |
| 1 | 2-Butenal | 1051 | 1047 | floral | 25 | 125 | - | MS, LRI, O |
| 2 | *trans*-2-Pentenal | 1137 | 1127 | fruity, green | - | 125 | - | MS, LRI, O |
| 3 | Butanol | 1141 | 1142 | sweet, whiskey | - | - | 25 | MS, LRI, O |
| 4 | 1-Penten-3-ol | 1155 | 1159 | horseradish, green | - | 125 | - | MS, LRI, O |
| 5 | 3-Methyl-2-butenal | 1202 | 1215 | fruity, sweet | - | 625 | - | MS, LRI, O |
| 6 | *trans*-2-Hexenal | 1224 | 1216 | green | - | 5 | - | MS, LRI, O |
| 7 | 3-Methyl pyridine | 1292 | 1292 | green, earthy | 25 | 5 | 5 | MS, LRI, O |
| 8 | *trans*-2-Heptenal | 1330 | 1323 | Vegetable, green | - | 25 | - | MS, LRI, O |
| 9 | *cis*-3-Hexenol | 1380 | 1382 | green, cut grass | 5 | 5 | - | MS, LRI, O |
| 10 | *trans*-2-Octenal | 1433 | 1429 | waxy, green | - | 125 | - | MS, LRI, O |
| 11 | Heptanol | 1448 | 1453 | herbal, green | 5 | 5 | - | MS, LRI, O |
| 12 | Furfural | 1469 | 1461 | sweet, woody | - | - | 25 | MS, LRI, O |
| 13 | *trans,trans*-2,4-Heptadienal | 1501 | 1495 | vegetable, green | - | 5 | - | MS, LRI, O |
| 14 | 1-(2-Furanyl)-ethanone | 1512 | 1499 | sweet, caramel | - | 25 | - | MS, LRI, O |
| 15 | Benzaldehyde | 1537 | 1520 | bitter almond, burnt sugar | 625 | 5 | - | MS, LRI, O |
| 16 | Linalool | 1537 | 1547 | floral, green | - | 25 | - | MS, LRI, O |
| 17 | Octanol | 1549 | 1557 | waxy, green | 5 | - | 5 | MS, LRI, O |
| 18 | *trans,cis*-2,6-Nonadienal | 1584 | 1584 | green, cucumber | - | - | 5 | MS, LRI, O |
| 19 | *γ*-Butanolactone | 1637 | 1632 | caramel, sweet | 5 | - | 5 | MS, LRI, O |
| 20 | 1-Nonanol | 1646 | 1660 | floral, rose | - | 5 | - | MS, LRI, O |
| 21 | *α*-Terpineol | 1696 | 1697 | woody, floral | 25 | 25 | 5 | MS, LRI, O |
| 22 | *γ*-Hexanolactone | 1723 | 1694 | sweet, herbal | 125 | 125 | 25 | MS, LRI, O |
| 23 | 1-Phenethyl alcohol | 1801 | 1801 | fresh, sweet | 5 | - | 5 | MS, LRI, O |
| 24 | Hexanoic acid | 1840 | 1846 | sour, cheese, fatty | 5 | 5 | 1 | MS, LRI, O |
| 25 | Guaiacol | 1861 | 1861 | phenolic, smoky | 5 | - | 25 | MS, LRI, O |
| 26 | Benzyl alcohol | 1879 | 1870 | sweet, floral | 5 | 25 | - | MS, LRI, O |
| 27 | *γ*-Octalactone | 1930 | 1910 | sweet, waxy | - | - | 125 | MS, LRI, O |
| 28 | 2-Phenethyl alcohol | 1914 | 1906 | honey, rose | 5 | 25 | 5 | MS, LRI, O |
| 29 | *β*-Ionone | 1971 | 1971 | seaweed, floral | 125 | 625 | 25 | MS, LRI, O |
| 30 | Phenol | 2004 | 2000 | phenolic, plastic, rubbery | 25 | - | 5 | MS, LRI, O |
| 31 | Pantolactone | 2029 | 2029 | candy | 5 | 125 | 25 | MS, LRI, O |
| 32 | *γ*-Nonalactone | 2041 | 2024 | coconut, creamy | - | - | 5 | MS, LRI, O |
| 33 | 4-Ethyl phenol | 2179 | 2187 | phenolic, smoky | 625 | 125 | 125 | MS, LRI, O |
| 34 | 2-Methoxy-4-vinyl phenol | 2197 | 2212 | woody | 5 | 25 | 25 | MS, LRI, O |
| 35 | Decanoic acid | 2262 | 2276 | sour, citrus | 625 | - | - | MS, LRI, O |
| 36 | Vanillin | 2578 | 2568 | vanilla, sweet | 625 | 125 | 125 | MS, LRI, O |

'-' Means compounds were not detected.

[†]Expt. LRI: linear retention index on an HP-Innowax column relative to C7-C40 alkane standards.

[‡]Ref. LRI: Reference retention index values from literature: NIST 14 MS library.

[§]Odor quality of compounds described by flavourists.

[‖]Flavour dilution-factor refers to the highest dilution at which the compound can be detected by at least three flavourists.

[⊥]Identification methods: MS = Comparison with mass spectrum of the compound in the NIST library version 2.2; LRI = Comparison of retention index with that of the compound in the NIST library version 2.2; O = Comparison of the retention time and odor quality of the eluted compound with standard.

not detected in LJF and LF. Besides that, *trans*-2-hexenal (green; FD-factor: 5), *trans*-2-heptenal (vegetable, green; FD-factor: 25), *trans*,*trans*-2,4-heptadienal (vegetable, green; FD-factor: 5), 1-(2-furanyl)-ethanone (sweet, caramel; FD-factor: 25), linalool (floral, green; FD-factor: 25) and 1-nonanol (floral, rose; FD-factor: 5) were only detected in LF, while butanol (sweet, whiskey; FD-factor: 25), furfural (sweet, woody; FD-factor: 25) and *trans*, *trans*-2,6-nonadienal (green, cucumber; FD-factor: 5) and γ-nonalactone (coconut, creamy; FD-factor: 5) were unique in LJC. In particular, pantolactone, which has a candy-like odor, could be confidently identified by GC-O with FD-factors of 5, 125, and 25 in LJF, LF, and LJC respectively, despite appearing as a trace compound in the GC-MS/FID analysis. The results suggested that the key odorants among LJF, LF, and LJC were quite different.

## Principal component analysis

Volatile studies often result in large, complicated datasets [35]. PCA is often used as a statistical tool to improve the efficiency of data analysis. In this study, PCA was conducted to pinpoint the main areas of variation among different species, parts and origins of honeysuckle. PCA could also suggest specific volatile features that contributed most significantly to the differences observed between honeysuckle samples. In the present study, the key odorants that were identified by GC-O and contributed to the aroma of honeysuckle were selected for comparison by PCA instead of the total volatile composition by GC-MS/FID. This allowed for a more streamlined analysis of various honeysuckles by eliminating volatile compounds that were not confirmed to be significant aroma contributors in honeysuckle. The data collected from dry and brewed LJF (Shandong and Henan province, China), LF (Hunan and Hubei province, China), LJC, and LC extracted by HS-SPME and SAFE are shown in Fig 4.

Fig 4(A) shows the biplot of key odorants in various honeysuckle extracted by HS-SPME-GC/MS without brewing. In the plot, PC 1 and 2 accounted for 45.6% and 27.0% of variations, respectively. Heptanol (11), 1-nonanol (20), and furfural (12) contributed greatly to the separation of LF from other species and parts of honeysuckle. Both LF samples extracted by HS-SPME contained more abundant amounts of these compounds regardless of their origins, although 1-nonanol (20) had a relatively greater peak area in LF Hunan. Similarly, γ-nonanolactone (32), 2-methoxy-4-vinyl phenol (34), and vanillin (36) were found to determine the separation of LJC on the positive area of PC 1 and negative area of PC 2 in the plot. In Fig 4 (A), the plot shows that the two LJF samples were clustered together, despite having different origins. A similar observation was seen in the two LF samples. Hence, the origin likely had a smaller impact on the differences in volatile compounds than the species. On the other hand, LJC and LC were separated from each other in the plot.

The comparison of key odorants in honeysuckle after brewing extracted by HS-SPME is shown in Fig 4(B). PC 1 and 2 accounted for 33.6% and 26.3% of the variations, respectively. In the plot, compounds, such as α-terpineol (21), furfural (12), 1-penten-3-ol (4) and octanol (17), determined the separation of LF from other species and parts of honeysuckle. α-Terpineol (21), furfural (12) and 1-penten-3-ol (4) were only detected in LF. Octanal (17) had a greater peak area in LF compared to other species. Similarly, hexanoic acid (24), γ-nonanolactone (32) and 2-methoxy-4-vinyl phenol (34) determined the plot on the positive area of PC 1 and PC 2. Hexanoic acid (24) shown a greater peak area in LJC. Similarly, to the dry samples, the separation was observed for different species, but not for different origins.

The results of key odorants of honeysuckle extracted by SAFE are shown in Fig 4(C). PC 1 was 34.5%, and PC 2 was 29.1%. In the plot, 1-(2-furanyl)-ethenone (14), linalool (16), 1-nonanol (20), and 3-methyl-2-butenal (5) determined the separation of LF on the positive area of PC 1 and the negative area of PC 2. 3-Methyl-2-butenal (5), 1-(2-furanyl)-ethenone (14),

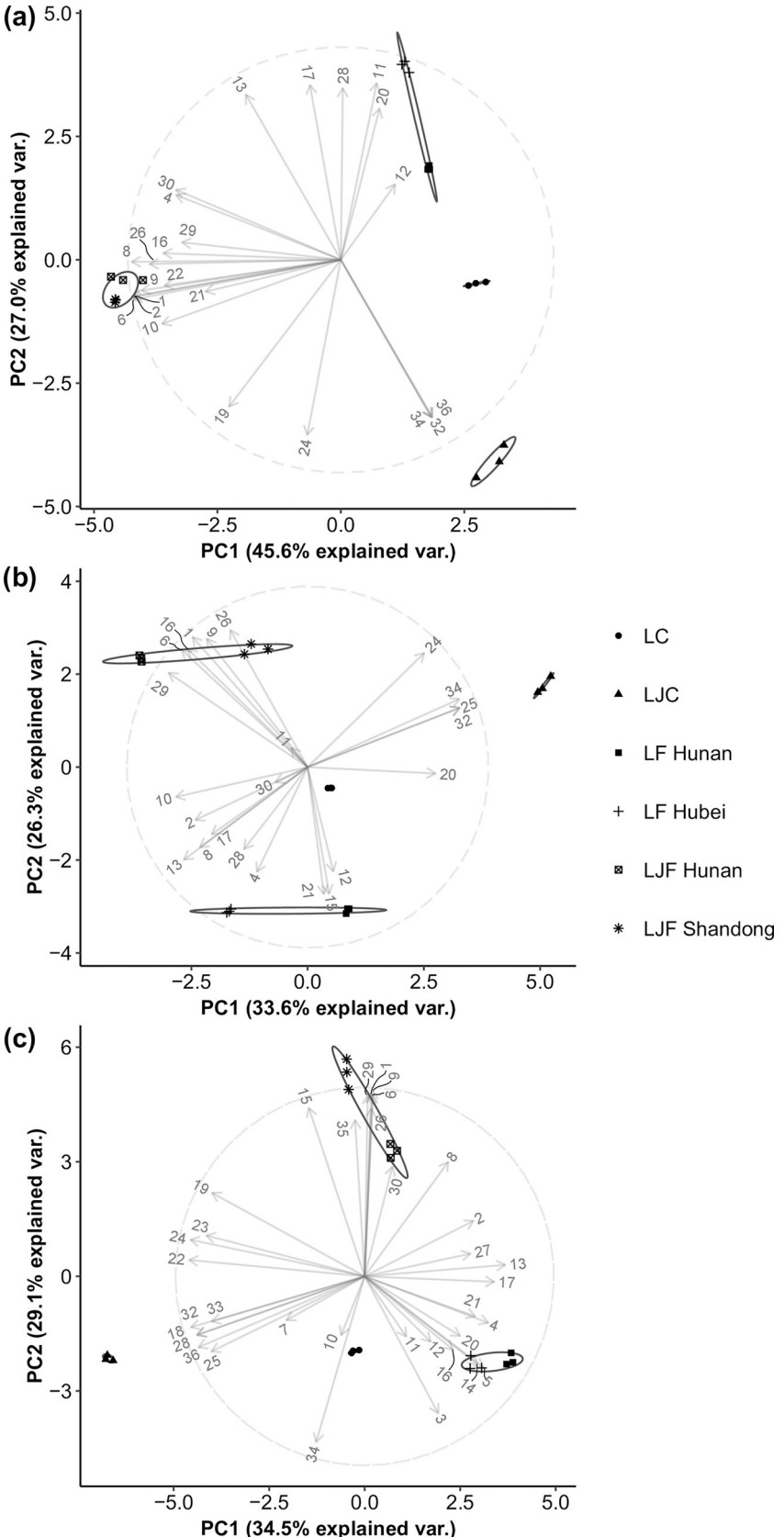

**Fig 4.** PCA biplot: (a) Dry honeysuckle samples using key odorants extracted by HS-SPME; (b) Brewed honeysuckle samples using key odorants extracted by HS-SPME; (c) Honeysuckle samples using key odorants extracted by SAFE; The numbers denote the corresponding volatiles reported in Table 1 for HS-SPME and SAFE loading plots.

linalool (16), and 1-nonanol (20) were only found in LF. Compounds, such as *trans*, *cis*-2,6-Nonadienal (18),) γ-nonalactone (32), 4-ethyl phenol (33), 2-methoxy-4-vinyl phenol (34) and vanillin (36), were found to determine LJC on the negative area of PC 1 and PC 2. *trans*, *cis*-2,6-Nonadienal (18) was only found in LJC, while the other four compounds had greater peak areas in LJC. Again, clustering was observed for LJF and LF of different origins. γ-Nonalactone (32) and 2-methoxy-4-vinyl phenol (34) determined the separation of LJC, and 1-nonanol (20) and furfural (12) determined the separation of LF from the other various honeysuckle extracted by HS-SPME and SAFE.

## Conclusion

In this research, it was found that HS-SPME was able to extract volatile aldehydes more efficiently than SAFE, with larger peak areas obtained in the dry flowers of honeysuckle than in the brewed samples. With the help of SAFE, volatile compounds were minimally modified, and the compounds of lower volatility (e.g., eugenol, decanoic acid, and vanillin) were significantly enhanced. Based on the AEDA results, 36 key odorants were identified across LJF, LF and LJC SAFE extracts. Compounds with the highest FD-factors were 4-ethyl phenol, decanoic acid, 3-methyl-2-butenal, β-ionone, γ-octalactone, and vanillin, and the concentrations of key odorants among different honeysuckle species and parts were noted to be quite varied. PCA classified various honeysuckle extracted by HS-SPME and SAFE by using key odorants, and clustering was observed for flowers of the same species regardless of origins. The results concluded that the key odorants may potentially be used to discriminate honeysuckle by species. Therefore, this systematic analytical approach is important for determining the volatile compounds of various honeysuckles and would definitely provide the basis for extending the application of honeysuckle in food products.

## Supporting information

**S1 Table. Identification of volatile compounds in dry and brewed *Lonicera japonica Flos*, *Lonicera japonica Caulis*, *Lonicera Flos*, and *Lonicera Caulis* extracted using HS-SPME (80˚C for 30 min).**
(DOCX)

**S2 Table. Identification of volatile compounds and concentration (ng/mL) in *Lonicera japonica Flos*, *Lonicera Flos*, *Lonicera japonica Caulis*, and *Lonicera Caulis* extracted using SAFE.**
(DOCX)

## Acknowledgments

The authors were grateful to Mane SEA Pte Ltd for providing technical assistance during the study. The authors are thankful to Martin Pelerentegui, Jenny Suwardi, Leong Kwong Chee and Carine Orecchioni for their effort and contribution to this project.

## Author Contributions

**Conceptualization:** Bin Yu.

**Methodology:** Bin Yu.

**Project administration:** Shao Quan Liu, Benjamin Lassabliere.

**Resources:** XiaoXue Hu.

**Writing – original draft:** Keran Su.

**Writing – review & editing:** Xin Zhang, LiHui Jia, Kim Huey Ee, Aileen Pua, Rui Min Vivian Goh, Jingcan Sun.

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
