## [Decision Letter · Decision Letter 0]

30 Jun 2020

PONE-D-20-11596

Identification of Key Odorounts in Honeysuckle by Headspace-Solid Phase Microextraction and Solvent-Assisted Flavour Evaporation with Gas Chromatography-Mass Spectrometry and Gas Chromatograph-Olfactometry in Combination with Chemometrics

PLOS ONE

Dear Dr. Bin Yu,

Thank you for submitting your manuscript to PLOS ONE. After careful consideration, we feel that it has merit but does not fully meet PLOS ONE’s publication criteria as it currently stands. Therefore, we invite you to submit a revised version of the manuscript that addresses the points raised during the review process.

We look forward to receiving your revised manuscript.

Kind regards,

Tommaso Lomonaco, Ph.D

Academic Editor

PLOS ONE

Journal Requirements:

2. Please can you provide the following information about the honeysuckle collection:

- the specific locations of the sites from which the honeysuckle samples were collected (including geographical coordinates)

- details of any permits/permissions obtained to collect the samples.

'The authors have declared that no competing interests exist' 

We note that one or more of the authors are employed by a commercial company: Mane SEA Pte Ltd.

5. Please include a copy of Table 3 which you refer to in your text on page 4.

Additional Editor Comments (if provided):

Dear Authors, the current version of the manuscript require major revisions.

Please address all the comments of the reviewers.

Best regards,

Tommaso Lomonaco

Reviewers' comments:

Reviewer's Responses to Questions

**Comments to the Author**

1. Is the manuscript technically sound, and do the data support the conclusions?

Reviewer #1: Yes

Reviewer #2: Yes

2. Has the statistical analysis been performed appropriately and rigorously? 

Reviewer #1: Yes

Reviewer #2: N/A

3. Have the authors made all data underlying the findings in their manuscript fully available?

Reviewer #1: Yes

Reviewer #2: Yes

4. Is the manuscript presented in an intelligible fashion and written in standard English?

Reviewer #1: Yes

Reviewer #2: No

5. Review Comments to the Author

Reviewer #1: The article is interesting, the experiment is well-planned and clearly described. The methods are adequately described; the choice of samples and components on chromatograms is justified. There are questions on statistical data processing: how many samples were investigated in parallel? Figures 1, 2 show very small ranges of variation for individual components, is this really so? In order for the intervals to be such, the number of parallel-studied samples should be large. If so, then the values in the statistical columns are significantly different and the corresponding p values should be calculated. Regarding PCA: it is not indicated whether the 2 main components are sufficient to separate the groups. Figures 3b are uninformative, it may be better to give a correlation circle or simply give the values of the correlation coefficients with the composition of the honeysuckle. The conclusions and annotations refer to classification, but no numerical characteristics of the classification are mentioned. It also seems to me that it would be appropriate to give an example of a chromatogram and give in the text of the article a table with the quantitative content of the determined components, since it is not very convenient to refer to additional materials. I believe that after making the appropriate changes, the article can be published.

Reviewer #2: Reviewer

In this paper, a full and systematic characterization of key odourants in honeysuckle is interestingly presented and discussed. The use of HS-SPME and SAFE in combination with GC-MS and AEDA with GC-O represents a valid approach for the chemical characterization of the volatile fraction of honeysuckle and the identification of the aroma, which is generally based on experienced tasters.

I would like to suggest some comments to improve the manuscript:

Main general comments:

i) Several sentences along the text are unclear or incorrect. Please improve your writing skills in English, thus restating the text more clearly.

ii) In the “Results and Discussion” section, the discussion is not extremely exhaustive. Please furnish a more comprehensive discussion of your data with respect to the literature.

Specific comments:

Abstract:

Line 29: Change “polarity” with “polar”

Line 34: “Finally, principal component analysis (PCA) analysed”. Unclear sentence, restate. PCA cannot be the subject of this sentence.

Line 35: In the sentence “Their results suggested that..”. What does “Their” refer to?

Line 38: Too vague.

Introduction:

Line 46: After reference [1] put a dot and start the following sentence as “It has been consumed…”

Line 47: Rephrase “In addition to treat pneumonia and respiratory infection and improve consumers’ health,..” as “In addition to the treatment of pneumonia and respiratory infection and the improvement of consumers' health,..”

Line 53: Rephrase “but the quality of honeysuckle is evaluated by professional tasters presently that..” as “. At present, the quality of honeysuckle is evaluated by professional tasters that..”

Line 59: Change “techniques” with “approaches”.

Line 66: Change “polarity” with “polar”

Line 75: Rephrase “Hence, the aromas of different honeysuckle species were difficult to distinguish..” as “ Hence, it was difficult to distinguish between the aromas of different honeysuckle species”

Line 82: Change “and analysed by GCMS” with “both coupled to GCMS”

Materials and methods:

Line 91: put a comma before “but”.

Line 98: “All samples were stored at 5 ℃ in airtight aluminium sachets until use.” How long? What about the influence of humidity? May be advisable the use of desiccants?

Line 102: “0.200 g of each honeysuckle species” do you refer to flowers, stem or both? Please specify in the text.

Line 104: If the numbers refer to nominal values, please express them as integer and change along the text.

Line 109: Specify the extraction conditions, i.e. time and temperature

Line 110: Change “followed by GC analysis” with “for the subsequent GC analysis”

Line 112: Rephrase “but the sample size was increased to 60.000 g and 600.0 g of Ultrapure water was used” as “but in this case 60 g of sample and 600 g of Ultrapure water were used”

Line 113: Internal standard solution (0.060 g). Does this weight correspond to 50 mL of the solution prepared in ethanol?

Line 114: “was added to 300 mL of filtrate” Is it the chilled filtrate? Where exactly was the filtrate contained? Which was the container? Please add this information.

Line 115: Why choosing butyl butyryl lactate as internal standard?

Line 135: What does “Mean values” refer to? Peak area? Peak intensity? Peak area corrected with respect to internal standard (if used)?

Line 141: Add a brief description of FD-factor.

Line 143: What do you mean exactly with odour quality?

Lines 146-148: Restate the sentence more clearly

Line 149: Why using the Benjamini-Hochberg Multiple Testing Correction?

Line 152: Rephrase “entity list were used” as “entity list and were used”.

Results and discussion

Line 157: Change “lead” with “could lead”

Line 158: Add a reference after “recovery.”

Line 160: Rephrase “volatile compounds of honeysuckle (dry and brewed)” as “volatile compounds from honeysuckle (both dry and brewed)”

Lines 165-168: Why did you make a list of five compounds as indicators and then you discussed the results with respect to some others, e.g. (e.g. 2-butenal, heptanal, trans-2-hexenal, benzaldehyde, geraniol, octanol, and 2-phenylethyl alcohol)? Please, restate sentence at line 161 and describe here the overall panel of indicators, with more or less 3-4 compounds for each chemical class.

Line 170: Hexanal or hexenal? Please correct.

Line 174: “higher peak areas” How much higher? Be specific and express in percentage with respect to the temperature.

Line 178: Rephrase “trends as peak areas increasing with extraction temperature increase after 60 °C” as “as the peak areas increase when the extraction temperature is above 60°C”.

Line 182: “various extraction times at 80 °C” Did you work one factor at a time? Why did you not use experimental design for the optimization of the extraction step?

Line 185: Largest??? What do you mean? Please change.

Line 187: “the peak area of hexanal first decreased and then increased” Be specific on the timeframe of decrease/increase.

Line 189: When referring to LJC and LJF, do you mean dry LJC and LJF? Please specify.

Line 190: “Because” is not required. Please change.

Line 193: Please add “both in dry and brewed samples” ate the end of the sentence.

Line 197: Tentatively identified? Did you check with pure standards? In some cases you did not, thus it is just a tentative identification. Please add this information

Line 198: Rephrase “were preferential” as “ were preferentially found”

Line 199: “identified”. This was true, only if the retention time and MS spectrum was matched with that of the corresponding pure standard. Some of the compounds reported in table were just tentatively identified.

Line 199: In S1 Table, does the mean ± dev std refer to the peak area? The peak area of the full scan or EIC chromatogram? If EIC, please specified the selected ion in the table?

Line 203: Restate the sentence more clearly

Line 207: “This might be due to different extraction methods..” Explain better. Which are the main differences between your extraction procedure and the common one?

Line 210: “they only focused on one species”. On which one? Be specific

Line 216: In S2 Table, does the mean value refer to a mean concentration? Explain more clearly in the materials and methods section how exactly it was obtained (preparation of calibration curves) and calculated

How did you obtained the concentration of compounds whose pure standards were not available in lab?

Did you normalize the data to the internal standard to control the analyte variations across the entire analytical procedure? If you did, please specify. If you did not, how could you assess the reliability of your experiments?

Line 217: Please insert in the text just few examples of the compounds listed in S2 Table and the agreement with chemicals reported the literature

Line 218: Change “polarity” with “polar”

Line 243: Not detected means a FD below XXX? Please specify.

Lines 273-274: “to pinpoint the main areas of variation among different species, parts and origins of honeysuckle in this study.” Please restate as a separate sentence.

Lines 275-276: Rephrase “In this present study, key odourants were identified by GC-O, which contributed to the aroma of honeysuckle” as “In the present study, the key odourants that were identified by GC-O and contributed to the aroma of honeysuckle”

Line277: Among the HS-SPME-GC/MS AND SAFE-GC/MS data, did you just select the compounds identified through GC-O? Compounds identified through the match with standard or also tentatively identified? Explain more clearly.

Line 287: Larger??? Please change.

Line 289: In which direction of the plot?

Line 293: In which direction? In correspondence of which variable?

Line 298: In which direction? Be specific

Line 299: Larger??? Please change.

Conclusion

Line 327: “approach is critical”. Please change “critical”. It has a negative meaning.

6. PLOS authors have the option to publish the peer review history of their article (what does this mean?). If published, this will include your full peer review and any attached files.

Reviewer #1: No

Reviewer #2: No

---

## [Author Response · Author response to Decision Letter 0]

22 Jul 2020

Reference Number: PONE-D-20-11596

Dear Editor and Reviewers,

We wish to thank you for the time and effort spent on reviewing our manuscript. Motivated by your extensive and insightful comments, we have modified the manuscript and hope that the revised version will be reconsidered. Our responses to your questions, comments and suggestions (repeated below for your convenience) are as follows:

Editor’s comments:

1. Please can you provide the following information about the honeysuckle collection:

- the specific locations of the sites from which the honeysuckle samples were collected (including geographical coordinates)

Authors’ answer:

Thank you for your suggestion. I have added the specific locations of the honeysuckle samples in the section of sample preparation.

- details of any permits/permissions obtained to collect the samples.

Authors’ answer:

Thank you for your suggestion. I have provided the permissions in the section of sample preparation.

2. Please include a copy of Table 3 which you refer to in your text on page 4.

Authors’ answer:

Thank you for your suggestion. I have corrected the error and changed the Table 3 with Table 1.

Reviewer #1’s comments:

General comments: 

The article is interesting, the experiment is well-planned and clearly described. The methods are adequately described; the choice of samples and components on chromatograms is justified. 

Authors’ answer: 

Thank you for your affirmation of our work.

1. There are questions on statistical data processing: how many samples were investigated in parallel? Figures 1, 2 show very small ranges of variation for individual components, is this really so? In order for the intervals to be such, the number of parallel-studied samples should be large. If so, then the values in the statistical columns are significantly different and the corresponding p values should be calculated.

Authors’ answer: 

Thanks for your question. Every sample was carried out in triplicate at each extraction condition. Among the six selected compounds in Figures 1 and 2, the compound with the greatest peak area was about 106 times larger than the compound with the smallest peak area. Hence, the log scale was chosen as the ordinate to show the trends of all six compounds clearly on the coordinate axis. Given your insight, we have changed the coordinates of Figures 1 and 2 for greater ease of understanding. As you suggested, the p-values have also been added in the figures.

2. Regarding PCA: it is not indicated whether the 2 main components are sufficient to separate the groups. Figures 3b are uninformative, it may be better to give a correlation circle or simply give the values of the correlation coefficients with the composition of the honeysuckle. The conclusions and annotations refer to classification, but no numerical characteristics of the classification are mentioned. 

Authors’ answer:

PC 1 and PC 2 account for the vast majority of the variation. This indicates that a 2-D graph, using just PC 1 and PC 2, would be a sufficient representation of significant areas of variation and it would account for a considerable 72.6 % (Figure 4(a)), 59.9 % (Figure 4(b)), and 63.6 % (Figure 4(c)) of the total variation in the data. In addition, we noted that the samples in the 2-D graph can be well separated, and areas of variation are clear (as opposed to the 3-D graph). As suggested, we replotted the PCA by R programming language for added detail, and the correlation circle was added in the biplot.

3. It also seems to me that it would be appropriate to give an example of a chromatogram and give in the text of the article a table with the quantitative content of the determined components, since it is not very convenient to refer to additional materials.

Authors’ answer:

Thanks for your suggestion. The chromatogram has been added in the manuscript.

Reviewer #2’s comments:

General comments: 

In this paper, a full and systematic characterization of key odourants in honeysuckle is interestingly presented and discussed. The use of HS-SPME and SAFE in combination with GC-MS and AEDA with GC-O represents a valid approach for the chemical characterization of the volatile fraction of honeysuckle and the identification of the aroma, which is generally based on experienced tasters.

Authors’ answer: 

Thank you for your affirmation of our work.

Main general comments:

i) Several sentences along the text are unclear or incorrect. Please improve your writing skills in English, thus restating the text more clearly.

Authors’ answer:

Thanks for your suggestion. We have corrected the unclear and incorrect sentences in the manuscript.

ii) In the “Results and Discussion” section, the discussion is not extremely exhaustive. Please furnish a more comprehensive discussion of your data with respect to the literature.

Authors’ answer:

Thanks for your suggestion. We have furnished the discussion of the results based on the corresponding references.

Specific comments:

Abstract:

1. Line 29: Change “polarity” with “polar”

Authors’ answer: 

Thank you for your correction. We have corrected the word error on line 30.

2. Line 34: “Finally, principal component analysis (PCA) analysed”. Unclear sentence, restate. PCA cannot be the subject of this sentence.

Authors’ answer: 

Thank you for pointing out our mistakes. We have corrected the unclear sentence on line 35.

3. Line 35: In the sentence “Their results suggested that..”. What does “Their” refer to?

Authors’ answer: 

Thanks for your correction. We have changed “Their” with specific information on line 36.

4. Line 38: Too vague.

Authors’ answer: 

Sorry to make you confused and thank you for your question. We have corrected the sentence on line 39.

Introduction:

1. Line 46: After reference [1] put a dot and start the following sentence as “It has been consumed…”

Authors’ answer: 

Thanks for your suggestion. We have added a dot on line 48 after reference [1].

2. Line 47: Rephrase “In addition to treat pneumonia and respiratory infection and improve consumers’ health,..” as “In addition to the treatment of pneumonia and respiratory infection and the improvement of consumers' health,..”

Authors’ answer: 

Thank you for pointing out our mistakes. We have corrected the sentence on line 49 and 50.

3. Line 53: Rephrase “but the quality of honeysuckle is evaluated by professional tasters presently that..” as “. At present, the quality of honeysuckle is evaluated by professional tasters that..”

Authors’ answer: 

Thanks for your correction. We have corrected the sentence on line 55 and 56.

4. Line 59: Change “techniques” with “approaches”.

Authors’ answer: 

Thank you for your suggestion. We have changed “techniques” with “approaches” on line 61.

5. Line 66: Change “polarity” with “polar”

Authors’ answer: 

Thanks for your correction. We have changed “polarity” with “polar” on line 68.

6. Line 75: Rephrase “Hence, the aromas of different honeysuckle species were difficult to distinguish.” as “ Hence, it was difficult to distinguish between the aromas of different honeysuckle species”

Authors’ answer: 

Thank you for pointing out our mistakes. We have corrected the sentence on line 77.

7. Line 82: Change “and analysed by GCMS” with “both coupled to GCMS”

Authors’ answer: 

Thank you for your correction. We have corrected the sentence on line 84.

Materials and methods:

1. Line 91: put a comma before “but”.

Authors’ answer: 

Thanks for your correction. We have added a comma before “but” on line 93.

2. Line 98: “All samples were stored at 5 ℃ in airtight aluminium sachets until use.” How long? What about the influence of humidity? May be advisable the use of desiccants?

Authors’ answer: 

Thank you for your question. One-kilogram honeysuckle samples were sealed in a package. After opening, every 100 g sample was put into a small heat-sealed vacuum aluminium foil bag and stored in the 5 ℃ refrigerator. The aluminium foil bag can isolate the outside humidity, so no desiccant is needed. All samples are used within one week after repacking. Once each bag of sample was opened, it will not be used again. We have added the information on line 96-105.

3. Line 102: “0.200 g of each honeysuckle species” do you refer to flowers, stem or both? Please specify in the text.

Authors’ answer: 

Thank you for your question. We have added the information on line 109. LJF (Shandong and Henan), LF (Huan and Hubei), LC, and LJC were all extracted by HS-SPME. Hence, 0.200 g of each sample was weighted and put in the headspace vial.

4. Line 104: If the numbers refer to nominal values, please express them as integer and change along the text.

Authors’ answer: 

Thanks for your suggestion. The measurement scale of the balance we used is accurate to three decimal places or one decimal place. Hence, 0.200 g has three decimal places. The information of balance has been added on line 109.

5. Line 109: Specify the extraction conditions, i.e. time and temperature

Authors’ answer: 

Thanks for your suggestion, the extraction time and temperature have been added in the manuscript on line 118 to 120.

6. Line 110: Change “followed by GC analysis” with “for the subsequent GC analysis”

Authors’ answer: 

Thank you for pointing out our mistakes. We have corrected the sentence on line 120. 

7. Line 112: Rephrase “but the sample size was increased to 60.000 g and 600.0 g of Ultrapure water was used” as “but in this case 60 g of sample and 600 g of Ultrapure water were used”

Authors’ answer: 

Thanks for your correction. We have corrected the sentence on line 123.

8. Line 113: Internal standard solution (0.060 g). Does this weight correspond to 50 mL of the solution prepared in ethanol? 

Authors’ answer: 

Sorry to make you confused, and we have rephrased this sentence for a better understanding. 0.300 g butyl butyryl lactate was added in 250 mL volumetric flask, then dilute to the desired volume with ethanol. This solution was then referred to as the internal standard. Then, 0.060 g internal standard solution was added to 300 mL filtrate sample solution.

9. Line 114: “was added to 300 mL of filtrate” Is it the chilled filtrate? Where exactly was the filtrate contained? Which was the container? Please add this information. 

Authors’ answer: 

Sorry to make you confused, and we have rephrased this sentence and added the following information in the manuscript for a better understanding. 60.000 g of honeysuckle was added in 600.0 g of 80 ℃ Ultrapure water. After brewing, the mixture was filtered through a metal sieve and chilled in an ice bath for 10 min. 300 mL of the chilled filtrate was then transferred into a 500 mL beaker. Internal standard solution (0.060 g) was added to the 300 mL of filtrate and stirred for 5 min.

10. Line 115: Why choosing butyl butyryl lactate as internal standard? 

Authors’ answer:

Thank you for your question. The peak of butyl butyryl lactate does not coelute with the other naturally-occurring compounds in honeysuckle (validated by overlaying the chromatograms for the internal standard, and for all samples of honeysuckle that were unspiked). In addition, the retention time of butyryl lactate is in the middle of all volatile compounds in honeysuckle based on our GC set-up and method applied. Hence, butyl butyryl lactate was chosen as internal standard.

11. Line 135: What does “Mean values” refer to? Peak area? Peak intensity? Peak area corrected with respect to internal standard (if used)?

Authors’ answer:

Thanks for your question. “Mean values” refer to the average peak area of each compound extracted by HS-SPME and average concentration of each compound extracted by SAFE. We have added the information under the S1 and S2 table.

12. Line 141: Add a brief description of FD-factor.

Authors’ answer:

Thanks for your suggestion. We have added the description of FD-factor in section 2.5 of the manuscript.

13. Line 143: What do you mean exactly with odour quality? 

Authors’ answer:

Thanks for your question. The odour quality means odour description, such as floral and fruity. We have changed the “odour quality” with “odour description” for easier understanding in the manuscript.

14. Lines 146-148: Restate the sentence more clearly

Authors’ answer:

Thanks for your suggestion. We have rewritten part of section 2.6 in order to improve the clarity of the content.

15. Line 149: Why using the Benjamini-Hochberg Multiple Testing Correction?

Authors’ answer:

Thanks for your question. We have changed the method of the calculation of PCA using R instead. The data was processed and transformed with data.table and magrittr packages and visualized them with ggplot2, ggbiplot, and patchwork packages.

16. Line 152: Rephrase “entity list were used” as “entity list and were used”.

Authors’ answer:

Thanks for your correction. We have rewritten the part of section 2.6.

Results and discussion

1. Line 157: Change “lead” with “could lead”

Authors’ answer:

Thanks for your correction. We have added “could” before “lead”.

2. Line 158: Add a reference after “recovery.” 

Authors’ answer:

Thanks for your suggestion. We have added the following references.

https://doi-org.libproxy1.nus.edu.sg/10.1111/j.1750-3841.2011.02474.x
https://doi.org/10.1016/j.foodres.2019.02.044

3. Line 160: Rephrase “volatile compounds of honeysuckle (dry and brewed)” as “volatile compounds from honeysuckle (both dry and brewed)”

Authors’ answer:

Thank you for pointing out our mistakes. We have corrected the sentence on line 187.

4. Lines 165-168: Why did you make a list of five compounds as indicators and then you discussed the results with respect to some others, e.g. (e.g. 2-butenal, heptanal, trans-2-hexenal, benzaldehyde, geraniol, octanol, and 2-phenylethyl alcohol)? Please, restate sentence at line 161 and describe here the overall panel of indicators, with more or less 3-4 compounds for each chemical class.

Authors’ answer:

The comparison and grouping were conducted based on the chemical functional groups of the compounds. For example, hexanal was chosen as the representative of aldehyde compounds. 2-Butenal, heptanal, trans-2-hexenal, and benzaldehyde all contain an aldehylic functional group and were observed to undergo similar trends as hexanal. Geraniol, octanol, and 2-phenylethyl alcohol all contain the hydroxyl group and had the same trend with cis-3-hexenol and benzyl alcohol.

5. Line 170: Hexanal or hexenal? Please correct.

Authors’ answer:

Thanks for your correction. We have changed “hexenal” with “hexanal” in the manuscript.

6. Line 174: “higher peak areas” How much higher? Be specific and express in percentage with respect to the temperature.

Authors’ answer:

Thanks for your suggestion. We have calculated the corresponding p-values and added them on the figure 1 and 2. We hope that this makes it easier to see the differences between different extraction conditions.

7. Line 178: Rephrase “trends as peak areas increasing with extraction temperature increase after 60 °C” as “as the peak areas increase when the extraction temperature is above 60°C”.

Authors’ answer:

Thanks for your suggestion. We have rephrased the sentence on line 202. 

8. Line 182: “various extraction times at 80 °C” Did you work one factor at a time? Why did you not use experimental design for the optimization of the extraction step?

Authors’ answer:

Thanks for your suggestion. It is more reasonable to use an experimental design for the optimization of the extraction step. However, considering that the focus of this manuscript is to determine the key aroma compounds and distinguish different types of honeysuckle, we referred to the optimization conditions of other natural plants instead. This led us to select and optimise some specific extraction times and temperatures. It was finally decided to determine the most suitable extraction temperature and time from the extraction conditions mentioned in the manuscript. The reference is shown below.

https://doi.org/10.1016/j.lwt.2018.04.058

9. Line 185: Largest??? What do you mean? Please change.

Authors’ answer:

Thanks for your correction. We have changed “largest” with “greatest”.

10. Line 187: “the peak area of hexanal first decreased and then increased” Be specific on the timeframe of decrease/increase.

Authors’ answer:

Thanks for your suggestion. We have specified the information in the manuscript.

11. Line 189: When referring to LJC and LJF, do you mean dry LJC and LJF? Please specify.

Authors’ answer:

Thanks for your question. We have specified the information on line 221 in the manuscript.

12. Line 190: “Because” is not required. Please change.

Authors’ answer:

Thanks for your suggestion. We have deleted “Because” on line 222.

13. Line 193: Please add “both in dry and brewed samples” ate the end of the sentence.

Authors’ answer:

Thanks for your suggestion. We have added the information on line 225.

14. Line 197: Tentatively identified? Did you check with pure standards? In some cases you did not, thus it is just a tentative identification. Please add this information

Authors’ answer:

Thanks for your suggestion. We have added the information on line 239 in the manuscript. 

15. Line 198: Rephrase “were preferential” as “ were preferentially found”

Authors’ answer:

Thanks for your suggestion. We have rephrased the sentence on line 240.

16. Line 199: “identified”. This was true, only if the retention time and MS spectrum was matched with that of the corresponding pure standard. Some of the compounds reported in table were just tentatively identified.

Authors’ answer:

Thanks for your suggestion. We have replaced “identified compounds” by “results” on line 241.

17. Line 199: In S1 Table, does the mean ± dev std refer to the peak area? The peak area of the full scan or EIC chromatogram? If EIC, please specified the selected ion in the table?

Authors’ answer:

Thanks for your question. The mean ± dev std refers to the average peak area of each compound. Volatile compounds in honeysuckle were detected by GC-MS/FID. Hence, the peak area was obtained from FID detector.

18. Line 203: Restate the sentence more clearly

Authors’ answer:

Thanks for your suggestion. We have restated the sentence on line 246 to 248.

19. Line 207: “This might be due to different extraction methods..” Explain better. Which are the main differences between your extraction procedure and the common one?

Authors’ answer:

Thanks for your suggestion. We have specified the information on line 252.

20. Line 210: “they only focused on one species”. On which one? Be specific

Authors’ answer:

Thanks for your suggestion. We have specified the information on line 257.

21. Line 216: In S2 Table, does the mean value refer to a mean concentration? Explain more clearly in the materials and methods section how exactly it was obtained (preparation of calibration curves) and calculated

How did you obtained the concentration of compounds whose pure standards were not available in lab? 

Did you normalize the data to the internal standard to control the analyte variations across the entire analytical procedure? If you did, please specify. If you did not, how could you assess the reliability of your experiments?

Authors’ answer

Thank you for your question. We have added the explanation in the materials and methods.

The calculation of the concentration of compounds based on the formula that the ratio of the peak area of internal standard to its concentration is equal to the ratio of the peak area of each compound to the concentration of the corresponding compound. In this formula only the concentration of the compound is unknown. Hence, we utilised the internal standard peak ratio to semi-quantify the other volatile compounds present. 

As the concentration of each compound was further utilised when using PCA to compare different types of honeysuckle, these compounds were additionally verified by pure standards and GC-O simultaneously. Hence, the results are reliable.

22. Line 217: Please insert in the text just few examples of the compounds listed in S2 Table and the agreement with chemicals reported the literature

Authors’ answer

Thanks for your suggestion. We have added the compounds listed in S2 Table in the manuscript. To our best knowledge, this is the first time that volatile compounds were extracted by SAFE from honeysuckle. The results were well-corroborated as most of the volatile compounds extracted by SAFE are similar to those extracted by other extraction methods such as HS-SPME and SDE. The results were obtained by comparing S1 table and S2 table and referring to literature.

23. Line 218: Change “polarity” with “polar”

Authors’ answer

Thanks for your correction. We have changed “polarity” with “polar” on line 271.

24. Line 243: Not detected means a FD below XXX? Please specify.

Authors’ answer

Thanks for your suggestion. The FD factor of decanoic acid is 625 in LJF but this volatile compound could not be smelled by panellists even in the dilution of 50 and it cannot be detected by GC-MS. Hence, decanoic acid was not detected in LJC and LF.

25. Lines 273-274: “to pinpoint the main areas of variation among different species, parts and origins of honeysuckle in this study.” Please restate as a separate sentence.

Lines 275-276: Rephrase “In this present study, key odourants were identified by GC-O, which contributed to the aroma of honeysuckle” as “In the present study, the key odourants that were identified by GC-O and contributed to the aroma of honeysuckle”

Authors’ answer

Thanks for your suggestion. We have corrected the two sentences as suggested on line 325 and 328, respectively.

26. Line277: Among the HS-SPME-GC/MS AND SAFE-GC/MS data, did you just select the compounds identified through GC-O? Compounds identified through the match with standard or also tentatively identified? Explain more clearly.

Authors’ answer

Thanks for your question. Panellists described the odour of each compound and indicated the retention time of the odour by GC-O. The name of these compounds was identified by injection of pure standards. Hence, these key odourants we selected were identified through GC-O, MS, LRI and pure standards. We have added the information in the materials and method based on your question.

27. Line 287: Larger??? Please change.

Authors’ answer:

Thanks for your correction. We have change “larger” with “greater” on line 343.

28. Line 289: In which direction of the plot?

Line 293: In which direction? In correspondence of which variable?

Line 298: In which direction? Be specific

Authors’ answer:

Thanks for your question. We have added the information on line 345, 350, and 356.

29. Line 299: Larger??? Please change.

Authors’ answer:

Thanks for your correction. We have change “larger” with “greater” on line 357.

Conclusion

Line 327: “approach is critical”. Please change “critical”. It has a negative meaning.

Authors’ answer to conclusion:

Thanks for your suggestion. We have changed “critical” with “important” on line 395.

---

## [Decision Letter · Decision Letter 1]

5 Aug 2020

Identification of Key Odorounts in Honeysuckle by Headspace-Solid Phase Microextraction and Solvent-Assisted Flavour Evaporation with Gas Chromatography-Mass Spectrometry and Gas Chromatograph-Olfactometry in Combination with Chemometrics

PONE-D-20-11596R1

Dear Dr. Bin Yu,

We’re pleased to inform you that your manuscript has been judged scientifically suitable for publication and will be formally accepted for publication once it meets all outstanding technical requirements.

Kind regards,

Tommaso Lomonaco, Ph.D

Academic Editor

PLOS ONE

Dear Author, all the questions have been addressed and then I suggest to accept the article in the present form.

Regards,

Tommaso Lomonaco

Reviewers' comments:

Reviewer's Responses to Questions

**Comments to the Author**

1. If the authors have adequately addressed your comments raised in a previous round of review and you feel that this manuscript is now acceptable for publication, you may indicate that here to bypass the “Comments to the Author” section, enter your conflict of interest statement in the “Confidential to Editor” section, and submit your "Accept" recommendation.

Reviewer #1: (No Response)

Reviewer #2: All comments have been addressed

2. Is the manuscript technically sound, and do the data support the conclusions?

Reviewer #1: Yes

Reviewer #2: (No Response)

3. Has the statistical analysis been performed appropriately and rigorously? 

Reviewer #1: Yes

Reviewer #2: Yes

4. Have the authors made all data underlying the findings in their manuscript fully available?

Reviewer #1: Yes

Reviewer #2: Yes

5. Is the manuscript presented in an intelligible fashion and written in standard English?

Reviewer #1: Yes

Reviewer #2: Yes

6. Review Comments to the Author

Reviewer #1: The authors made all the changes in accordance with the recommendations of the reviewers, which significantly improved the article. In its present form, the article can be accepted for publication.

Reviewer #2: (No Response)

7. PLOS authors have the option to publish the peer review history of their article (what does this mean?). If published, this will include your full peer review and any attached files.

Reviewer #1: No

Reviewer #2: No

---

## [Editor Report · Acceptance letter]

10 Aug 2020

PONE-D-20-11596R1 

Identification of Key Odorants in Honeysuckle by Headspace-Solid Phase Microextraction and Solvent-Assisted Flavour Evaporation with Gas Chromatography-Mass Spectrometry and Gas Chromatograph-Olfactometry in Combination with Chemometrics 

Dear Dr. Yu:

I'm pleased to inform you that your manuscript has been deemed suitable for publication in PLOS ONE. Congratulations! Your manuscript is now with our production department. 

Kind regards, 

on behalf of

Dr. Tommaso Lomonaco 

Academic Editor

PLOS ONE